# Exploring failure of antimicrobial prophylaxis and pre-emptive therapy for transplant recipients: a systematic review

Anne-Grete Märtson ,[1] Martijn Bakker,[2] Hans Blokzijl,[3] Erik A M Verschuuren,[4] Stefan P Berger,[5] Lambert F R Span,[2] Tjip S van der Werf,[4,5] Jan-Willem C Alffenaar [1,6]

For numbered affiliations see end of article.

**Correspondence to**
Jan-Willem C Alffenaar;
j.w.c.alffenaar@umcg.nl

## ABSTRACT

**Objectives** Infections remain a threat for solid organ and stem cell transplant recipients. Antimicrobial prophylaxis and pre-emptive therapy have improved survival of these patients; however, the failure rates of prophylaxis are not negligible. The aim of this systematic review is to explore the reasons behind failure of antimicrobial prophylaxis and pre-emptive therapy.

**Setting** This systematic review included prospective randomised controlled trials and prospective single-arm studies.

**Participants** The studies included were on prophylaxis and pre-emptive therapy of opportunistic infections in transplant recipients. Studies were included from databases MEDLINE, CENTRAL and Embase published until October first 2018.

**Primary and secondary outcome measures** Primary outcome measures were breakthrough infections, adverse events leading to stopping of treatment, switching medication or dose reduction. Secondary outcome measures were acquired resistance to antimicrobials, antifungals or antivirals and death.

**Results** From 3317 identified records, 30 records from 24 studies with 2851 patients were included in the systematic review. Seventeen focused on prophylactic and pre-emptive treatment of cytomegalovirus and seven studies on invasive fungal infection. The main reasons for failure of prophylaxis and pre-emptive therapy were adverse events and breakthrough infections, which were described in 54% (13 studies) and 38% (9 studies) of the included studies, respectively. In 25%, six of the studies, a detailed description of patients who experienced failure of prophylaxis or pre-emptive therapy was unclear or lacking.

**Conclusions** Our results show that although failure is reported in the studies, the level of detail prohibits a detailed analysis of failure of prophylaxis and pre-emptive therapy. Clearly reporting on patients with a negative outcome should be improved. We have provided guidance on how to detect failure early in a clinical setting in accordance to the results from this systematic review.

**PROSPERO registration number** CRD42017077606.

### Strengths and limitations of this study

► To reduce selection bias, all the studies were independently reviewed and risk of bias was assessed by two authors and disagreements solved by including a third reviewer.

► One limitation of this systematic review is that the included studies were recruiting only adult patients.

► Inclusion of single-arm studies could be a potential limitation as these can cause bias in the systematic review.

► To reach a broad scope for the systematic review three databases, MEDLINE, Embase and CENTRAL, were searched.

► The systematic review was reported according to Preferred Reporting Items for Systematic Reviews and Meta-Analyses guidelines.

## INTRODUCTION

In spite of novel immunosuppressive regimens and antimicrobial prophylaxis, infectious complications remain a threat for solid organ and stem cell transplant (SCT) recipients.[1–5] These patients are especially susceptible to opportunistic infections like cytomegalovirus (CMV), *Pneumocystis jirovecii* pneumonia (PCP), febrile neutropoenia, human herpesvirus 6 (HHV-6) and invasive fungal infections (IFI).[1–5] Graft failure is a major risk of these opportunistic infections.[6–8] In recent years, organ transplantation and immunosuppressive regimes have developed greatly and thus become available for a wider patient population. This requires adequate antimicrobial prophylaxis guidelines and studies supporting the scientific evidence.

Antimicrobial prophylaxis and pre-emptive therapy are used as preventive measures; however, these vary notably among different transplant centres.[9 10] This can be explained by differences in local setting and lack of

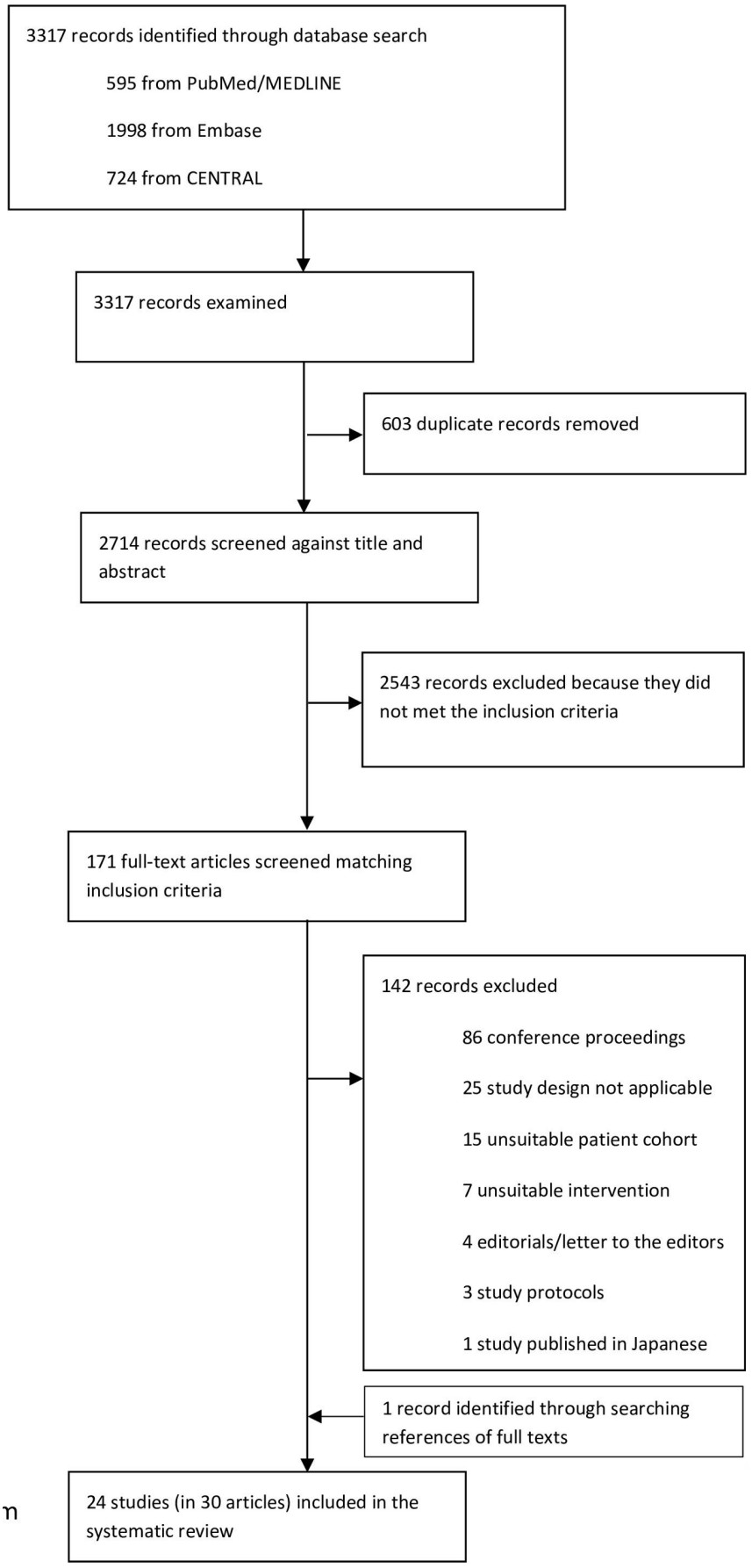

**Figure 1** PRISMA diagram. PRISMA, Preferred Reporting Items for Systematic Reviews and Meta-Analyses.+++

**Table 1** Study characteristics

CYTOMEGALVIRUS PROPHYLAXIS/PRE-EMPTIVE THERAPY

| Author, year | Country | Study design | Population | Pre-emptive therapy/prophylaxis | Prophylaxis/pre-emptive medication | Comparison | Sample size | Duration | Follow-up | Summary assessment of risk of bias |
|---|---|---|---|---|---|---|---|---|---|---|
| Humar et al, 2010[15 24] | Australia, Belgium, Canada, Brazil, France, Germany, the UK, Italy, Poland, Romania, Spain, the USA | RCT | Kidney transplant recipients aged 17–77 years | Prophylaxis | Valganciclovir | Valganciclovir different duration and placebo | 320 patients | 100 and 200 days | 12 months | Low risk |
| Halim et al, 2016[31] Ghei th et al, 2017[32] | Kuwait, Egypt | RCT | Kidney transplant recipients with mean age of 42.4 years | Prophylaxis | Valganciclovir | Valganciclovir different duration | 201 patients | 98.1 days (mean) | 12 months | Some concerns |
| Witzke et al, 2012[33], 2018[34] | Germany, Austria | RCT | Kidney transplant recipients with mean age of 52.7 years | Prophylaxis and pre-emptive therapy | Valganciclovir | Valganciclovir as pre-emptive treatment | 296 patients | 100 days (14 days pre-emptive until 100 days) | 7 years | Some concerns |
| Reischig et al, 2012[35] | Czeck Republic | RCT | Kidney transplant recipients with mean age of 49 years | Prophylaxis and pre-emptive therapy | Valganciclovir | Valacyclovir | 70 patients | 3 months | 2 years | Some concerns |
| Reischig et al, 2015[36] | Czeck Republic | RCT | Kidney transplant recipients with mean age of 49 years | Prophylaxis | Valganciclovir | Valacyclovir | 119 patients | 3 months | 1 year | Some concerns |
| Togashi et al, 2011[38] | Japan | RCT | Liver transplant recipients with mean age of 52.1 years | Pre-emptive therapy | Valganciclovir | Ganciclovir | 22 patients | 2 weeks | 1 year | Some concerns |
| Padulles et al, 2016[25] | Spain | RCT | Kidney, liver and heart transplant recipients with mean age of 53.9 years | Prophylaxis and treatment | Valganciclovir/ganciclovir | Valganciclovir/ganciclovir different dosing | 53 patients | NA (not reported) | 6 months | Low risk |
| Palmer et al, 2010, Finlen Copeland et al, 2011[26 27] | USA | RCT | Lung transplant recipients with median age of 55.5 years | Prophylaxis | Valganciclovir | Placebo | 136 patients | 12 months | 3.9 years (mean) | Some concerns |
| Chawla et al, 2011[28] | USA | RCT | Allogeneic transplant recipients aged 18–64 | Pre-emptive therapy | Valganciclovir | Ganciclovir | 37 patients | 28 days | 4 weeks | High risk |
| Boeckh et al, 2015[29] | USA | RCT | Allogeneic transplant recipients 16–70 | Pre-emptive therapy | Valganciclovir | Placebo | 184 patients | 150/120 days valganciclovir/placebo(medians) | 640 days | Some concerns |
| Kim et al, 2010[30] | South Korea | RCT | Allogeneic transplant recipients 16–49 | Pre-emptive therapy | Ganciclovir | Ganciclovir different dosing | 68 patients | 14 days (median) | 42 months (median) | Some concerns |
| Montejo et al, 2010[39] | Spain | Single-arm | Kidney transplant recipients with median age 47 years | Prophylaxis | Valganciclovir | NA | 34 patients | 3 months | 1 year | Critical risk |
| Nanmoku et al, 2018[40] | Japan | Single-arm | Kidney transplant recipients with mean age 48.7 years | Prophylaxis | Valganciclovir | NA | 100 patients | 6 months | 1 year | Serious risk |
| Perrottet et al, 2009 Manuel et al, 2010[9 41] | France | Single-arm | Kidney, heart, lung, liver transplant recipients aged 18–70 | Prophylaxis | Valganciclovir and ganciclovir | NA | 65 patients | 3 months | 1 year | Serious risk |

**Table 1** Continued

| Author, year | Country | Study design | Population | Pre-emptive therapy/prophylaxis | Prophylaxis/pre-emptive medication | Comparison | Sample size | Duration | Follow-up | Summary assessment of risk of bias |
|---|---|---|---|---|---|---|---|---|---|---|
| McGillicuddy et al, 2010 | USA | Single-arm | Kidney and pancreas transplant recipients mean age 53.5 years | Pre-emptive therapy and prophylaxis | Valganciclovir/ganciclovir | Valganciclovir/ganciclovir pre-emptive therapy | 130 patients | 90 days | 1 year | Serious risk |
| Takenaka et al, 2012[45] | Japan | Single-arm | Allogeneic transplant recipients aged 25–68 | Pre-emptive therapy | Valganciclovir | NA | 20 patients | 2 weeks | 10 weeks | Serious risk |
| Park et al, 2012[46] | South Korea | Single-arm | Allogeneic transplant recipients aged 16–70 | Pre-emptive therapy | Ganciclovir | Ganciclovir different dosing | 97 patients | 14 days (median) | Not same for all patients | Moderate risk |
| INVASIVE FUNGAL INFECTION PROPHYLAXIS | | | | | | | | | | |
| Perrella et al, 2012[47] | Italy | RCT | Liver transplant recipients (age NI) | Prophylaxis | Fluconazole | Amphotericin B | 43 patients | 7–14 days | 1 year | Some concerns |
| Winston et al, 2014[17] | USA | RCT | Liver transplant recipients aged 19–75 years | Prophylaxis | Fluconazole | Anidulafungin | 200 patients | 21 days (median) | 90 days | Low risk |
| Chaftari et al, 2012[48] | USA | RCT | Allogeneic transplant recipients aged 20–69 | Prophylaxis | Posaconazole | Amphotericin B lipid complex | 40 patients | 6 weeks | 2 weeks | Some concerns |
| Huang et al, 2012[19] | China | RCT | Allogeneic and autologous transplant recipients with mean age 32.7 years | Prophylaxis | Itraconazole | Micafungin | 228 patients | 42 days | 4 weeks | Some concerns |
| Park et al, 2016[49] | South Korea | RCT | Allogeneic and autologous transplant recipients aged 20–64 | Prophylaxis | Fluconazole | Micafungin | 250 patients | 21 days | 100 days | Some concerns |
| Mitsani et al, 2012[43] | USA | Single-arm | Lung transplant recipients aged 20–74 years | Prophylaxis | Voriconazole | NA | 93 patients | 3 months | NI | Moderate risk |
| Cordonnier et al, 2010[44] | France, Spain, Belgium, UK, Germany, Switzerland, Portugal, Sweden | Single-arm | Allogeneic transplant recipients aged 22–72 years | Prophylaxis | Voriconazole | NA | 45 patients | 94 days (median) | 1 year | Critical risk |

Summary assessment of risk of bias grades for:

RCTs using RoB tool:

Low risk of bias: The study is judged to be at low risk of bias for all domains for this result.

Some concerns: The study is judged to raise some concerns in at least one domain for this result, but not to be at high risk of bias for any domain.

High risk of bias: The study is judged to be at high risk of bias in at least one domain for this result.

Single-arm prospective studies using ROBINS tool::

Low risk of bias: The study is judged to be at low risk of bias for all domains.

Moderate risk of bias: The study is judged to be at low or moderate risk of bias for all domains.

Serious risk of bias: The study is judged to be at serious risk of bias in at least one domain, but not at critical risk of bias in any domain.

Critical risk of bias: The study is judged to be at critical risk of bias in at least one domain.

NA, not applicable; NI, no information; RCT, randomised controlled trial.

high level evidence in antifungal, antimicrobial and antiviral prophylaxis guidelines and fairly dated evidence for some infections like PCP and febrile neutropenia.[5 11–14] Although prophylaxis has proven to be a beneficial strategy, breakthrough infections and adverse events resulting in discontinuation of prophylaxis occur.[15–17] The rates of failure of prophylaxis have remained around 10%–20% for opportunistic infections.[15 18–20] In order to be able to optimise prophylaxis, it is important to understand underlying causes for failure. This systematic review will identify (1) causes of failure of prophylactic treatment, (2) factors that might contribute to failure of prophylaxis and (3) different approaches for administering prophylaxis.

This systematic review aimed to summarise the main reasons why prophylaxis and pre-emptive therapy has failed in solid organ and allogeneic SCT recipients and how failure is reported in prospective studies. In this review, we focus on failure of prophylactic therapy, during the treatment window.

## METHODS

### Definition of failure

In this systematic review, we have defined *failure of prophylaxis and pre-emptive therapy* as stopping or changing the therapy during the study period for any stated reason. For example, failure could be a breakthrough infection, non-adherence, adverse events leading to stopping of therapy, etc. Even if some of the side effects could be anticipated, if this leads to stopping of therapy, it was defined as failure. We did not look into infections in the postprophylactic period.

### Search strategy and selection criteria

Methods of the analysis and inclusion criteria were specified in advance and documented in a protocol, which is available online: https://www.crd.york.ac.uk/prospero/display_record.php?RecordID=77606 (online supplementary appendix 1). This report follows the Preferred Reporting Items for Systematic Reviews and Meta-Analyses (PRISMA) checklist (online supplementary appendix 2).[21] This systematic review included prospective randomised controlled trials (RCT) and prospective single-arm studies from 1 January 2010 to 1 October 2018. The starting date was 1 January 2010 due to the changes in management of different infections in recent decades and to include the most recent evidence.[11 12] There were no limitations for the patient setting, publication status and language. We analysed the failure during prophylaxis and pre-emptive therapy, thus the follow-up time varied. The review included patients (16 years and older) who had received either allogeneic stem cell, lung, kidney, liver, heart, pancreas or small bowel transplantation. Studies done on children under 16 years were not included as this would have introduced more variability and require a separate analysis, thus were out with the scope of this review. Moreover, these patients were receiving trimethoprim/sulfamethoxazole for prophylaxis of PCP, ciprofloxacin for prophylaxis of febrile neutropenia, ganciclovir and/or valganciclovir for prophylaxis or pre-emptive-therapy of CMV and human herpesvirus 6 (HHV-6); or posaconazole or voriconazole or fluconazole or itraconazole for prophylaxis of invasive fungal infections (IFI). Primary outcome measures were breakthrough infections, adverse events leading to stopping of treatment, switching medication or dose reduction. Secondary outcome measures were acquired resistance to antimicrobials, antifungals or antivirals and death.

To identify studies for this systematic review, the following databases were searched: MEDLINE (PubMed), EMBASE and Cochrane Central Register of Controlled Trials (CENTRAL). The search was performed on 1 October 2018. The search strategy included MeSH terms and variations of transplant types, medications and prophylaxis ('Antibiotic Prophylaxis' (Mesh) OR antimicrobial OR antimicrobial) AND (Host OR Transplants OR Transplantation OR Transplant Recipients OR immunocompromised OR transplant* OR kidney transplant*). The full-search strategies of all databases can be found in online supplementary appendix 3. The searches were done by AGM and MB. For screening, we used Covidence software (www.covidence.org).

The literature search and data extraction for inclusion and eligibility for this systematic review were done according to the inclusion criteria by AGM and MB independently. If there were discrepancies between the results, these were discussed and resolved with JWA or with consensus. If there was data missing or additional questions from the selected studies, then the authors of these studies were contacted. We excluded conference proceedings, retrospective studies, reviews, editorials and letters to the editor.

### Outcomes, data extraction and quality

Data were extracted by AGM and independently checked by MB. Disagreement between reviewers was resolved by discussion with a third reviewer JWA. Data were extracted from each included trial on: characteristics of patients, type of intervention, study design, study population, outcome measurement, reasons for failure of prophylaxis or pre-emptive therapy (stopping of prophylaxis/pre-emptive therapy), main conclusions by authors, strengths and limitations (online supplementary appendix 4).

We considered performing a meta-analysis for our systematic review, however because the patient cohorts include different transplantations and varied interventions, the studies were too heterogeneous, we decided to do a qualitative systematic review. The studies were divided into CMV and IFI prophylaxis and pre-emptive treatment groups.

Risk of bias in individual studies was assessed independently by AGM and MB. For assessing bias in individual studies, Revised Cochrane risk-of-bias tool for randomised trials (RoB 2) was used for RCTs and Risk of Bias in

Non-randomised Studies of Interventions (ROBINS-I) was used for prospective single-arm studies.[22 23]

## RESULTS

The search identified 3317 records for inclusion in the review. In total, 603 duplicate records were removed, and 2543 records were excluded after screening of title and abstract. Full-text screening of 171 articles resulted in 24 studies (in 30 articles) (PRISMA flow diagram in figure 1) to be included in the final review. We decided to exclude one full text in Japanese as it would not likely change the outcome of our systematic review. We contacted corresponding authors of 24 studies (30 articles) for further information about reasons for preliminary stopping of prophylaxis and overall adherence to treatment, seven responded and three of those sent prespecified protocols; however, none provided additional information about failure of prophylaxis or pre-emptive treatment. The individual study characteristics and risk of bias are presented in table 1.

We identified 24 studies including 2851 patients. Seventeen studies (11 RCTs[15 24–38] and six single-arm[39–46]) with 1952 subjects focused on CMV prophylaxis and seven studies (5 RCTs[17 19 47–49] and 2 single-arm[43 44]) with 899 subjects focused on IFI prophylaxis.

Of the 17 CMV studies, eight included only valganciclovir, two only ganciclovir, five included both valganciclovir and ganciclovir, two valaciclovir and valganciclovir. From all 17 CMV studies, seven focused only on prophylaxis, six only on pre-emptive therapy, three both on prophylaxis and pre-emptive therapy and one on prophylaxis and therapy.

The IFI studies varied with regard to study medication and patient group. Three studies evaluated fluconazole (comparison amphotericin B, anidulafungin and micafungin), one posaconazole (comparison amphotericin B lipid complex), one itraconazole (comparison micafungin) and two single-arm studies had voriconazole as study medication.

### Failure of prophylaxis and pre-emptive therapy

No specific information about failure during prophylaxis or pre-emptive therapy was given in 25% (six studies: four being RCTs) of the included studies.[19 25 28 40 42 47 50] Four of these studies did record follow-up infections or long-term failure of prophylaxis therapy after cessation of prophylaxis[19 40 42 47] and for one RCT the time-point was not specified.[46]

The most common reasons for failure of CMV prophylaxis (1524 study subjects) and CMV pre-emptive therapy (428 study subjects) were adverse events[15 25 27 29 30 36 42 46] and breakthrough CMV.[15 29 41 42 46] For IFI prophylaxis (899 study subjects), it was adverse events[19 43 44 48 49] and IFI.[17 19 44 49] Overall, the adverse events and breakthrough infections were described in 54% (13 studies) and 38% (9 studies) of the studies respectively. In table 2, the reasons for stopping prophylaxis are described in detail.

The detailed information about failure in the CMV (1506 study subjects) and IFI (761 study subjects) RCT groups are summarised in figures 2 and 3.

From the secondary outcomes, death was reported more frequently—in 33% (n=8) of the studies, death was the reason for failure.[17 27 30 31 35 36 44 46] The secondary outcome resistance to antimicrobials, antivirals and antifungals was addressed in the introductions and discussions of the included studies, however not regarded as failure of therapy. Moreover, the presence and/or measurement of resistance to the study drug was described in two studies.[27 29] Boeckh *et al* report no resistance genes in the investigated patients and Palmer *et al* report one patient with known resistance to ganciclovir.

Not all identified cases of failure could be clarified even after contact with the authors of the studies. The reasons for stopping prophylaxis and pre-emptive therapy, *patient reasons* or *physician or sponsor decision*,[27 29] were not explained in any of these studies. Moreover, in two studies,[24 27] it was stated that prophylaxis was stopped because of *other* reasons. Adverse events, breakthrough infections and cause of death were mostly described in further detail in the included studies.[15–17 27 29–31 35 42–44 46 48 49] However, reasons 'other',[15 27] physician/investigator decision[17 27 29 49] and patients discretion[15] were grouped together in studies and not described in detail.

### Risk of bias across studies

Risk of bias was assessed using five domains for RCTs: randomisation, assignment and adherence to intervention, measurement of outcome, missing data and selection of the reported results. For single-arm/observational studies domains were used: confounding, selection of participants, classification of interventions, missing data, measurement of outcomes and selection of the reported result. We concluded that all 16 RCTs and six single-arm had low risk of bias, three had some concerns and two had high risk of bias. Risk of bias in RCTs and single-arm studies is presented in tables 3 and 4.

## DISCUSSION

We aimed to explore the reasons why prophylaxis and pre-emptive therapy failed in transplant recipients. Twenty-four studies were included into this systematic review. We concluded that the main reason for stopping prophylaxis was adverse events for both CMV and IFI prophylaxis and CMV pre-emptive therapy. We did not observe notable differences between the prophylaxis and pre-emptive therapy groups. This result was expected as cessation of ganciclovir therapy is often described due to debilitating side-effects, especially bone marrow suppression,[15] which is even more problematic in haematological patients.[28] Different antifungal agents have a diverse safety profile. Our results were also in line with the common side-effects of antifungals as we observed nausea and vomiting in azoles and nephrotoxicity in Amphotericin B as reasons for discontinuation of therapy.[43 46 48]

**Table 2** Reasons for stopping prophylaxis or pre-emptive therapy

| Study | AE | Breakthrough infection | Graft loss | Death | Patient decision | Physician/investigator decision | Other | Total number subjects |
|---|---|---|---|---|---|---|---|---|
| **Cytomegalovirus prophylaxis/pre-emptive therapy** | | | | | | | | |
| Humar et al, 2010[15] | 27 (8%) | 51 (16%) | | | 11 (3%) | | 14 (4%) | 320 |
| Halim et al, 2016[31] | | | 2 (1%) | 1 (0.5%) | | | | 201 |
| Reischig et al, 2012[35] | | | 10 (14%) | 2 (3%) | | | | 70 |
| Reischig et al, 2015[36][1] | 13 (11%) | | 4 (3%) | 3 (2.5%) | | | | 119 |
| Padulles et al, 2016[25] | 1 (2%) | | | | | | | 53 |
| Palmer et al, 2010[27] | 17 (12.5%) | | | 7 (5%) | 3 (2%) | 15 (11%) | 3 (2%) | 136 |
| Boeckh et al, 2015[29] | 44 (24%) | 35 (19%) | | | | 20 (11%) | | 184 |
| Kim et al, 2010[30] | 1 (1.5%) | | | 2 (3%) | | | | 68 |
| Montejo et al, 2010[39] | | 3 (9%) | | | | | | 34 |
| Perrottet et al, 2009[41] | | 3 (viremia) | | | | | | 65 |
| McGillicuddy et al, 2010[42] | 3 (2%) | 1 (1%) (2) | | | | | | 130 |
| Park et al, 2012[49][3] | 8 (8%) | 6 (6%) | | 37 (38%) | | | | 97 |
| **Invasive fungal infection prophylaxis** | | | | | | | | |
| Winston et al, 2014[17][4] Fluconazole | | 2 (2%) | | 1 (1%) | | 2 (2%) | | 100 (fluconazole) 200 (total) |
| Chattari et al, 2012[48] Posaconazole | 8 (38%) | | | | | | | 21 (posaconazole) 40 (total) |
| Huang et al, 2012[19] Itraconazole | 29 (20%) | 12 (8%) | | 1 (1%) | | 4 (3%) | | 147 (itraconazole) 228 (total) |
| Park et al, 2016[49] Fluconazole | 2 (2%) | 3 (3%) | | | | 3 (3%) | | 89 (fluconazole) 250 (total) |
| Mitsani et al, 2012[43] | 25 (27%) | | | | | | | 93 |
| Cordonnier et al, 2010[44] | 2 (4%) | 3* (7%) | | 11* (2%) | | | | 45 |
| **NI in the following trials (timepoints not specified)** | | | | | | | | |
| Nanmoku et al, 2018[40] | | | | | | | | 100 |
| Chawla et al, 2011[28] | | | | | | | | 37 |
| Witzke et al, 2012[33] | | | | | | | | 296 |
| Perrella et al, 2012[47] | | | | | | | | 43 |
| **All patients completed treatment protocol in the following trials** | | | | | | | | |
| Togashi et al, 2011[38] | | | | | | | | 22 |
| Takenaka et al, 2012[45] | | | | | | | | 20 |

1. One patient moved to another country.
2. No information on breakthrough infections during prophylaxis.
3. 12 patients had dosage reductions.
4. 36 patients were preliminarily stopped due to discharge from hospital.
*Time-point not confirmed.
AE, adverse event; CMV, cytomegalovirus; IFI, invasive fungal infection; NI, no information (not reported).

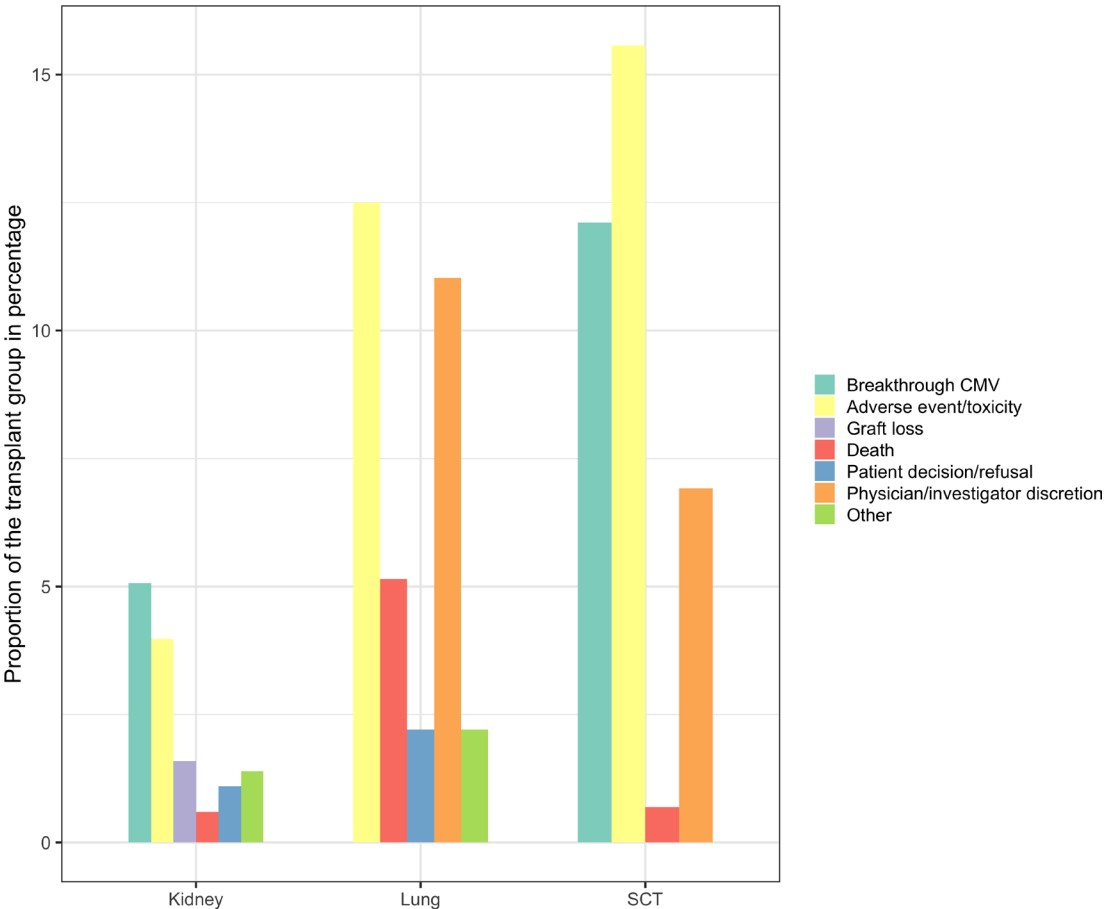

**Figure 2** Failure of CMV prophylaxis and pre-emptive therapy in 11 RCTs (y-axis represents the proportion of the transplant group in percentages and x-axis represents transplant). CMV, cytomegalovirus; RCTs, randomised controlled trials.

Adverse events are known to be under-reported so the numbers that we summarised in this systematic review might be underestimated.[51 52] A review by Golder *et al*[52] looked into different study designs and reporting of adverse events. The authors outlined that there is wide under-reporting of adverse events across different study designs and even more in the unpublished data. Unfortunately, contacting the authors of the studies included in this systematic review did not result in additional information about the reported adverse events. Meta-analysis was not done as the studies were too heterogeneous.

Surprisingly, reporting of failure was insufficient in 25% of RCTs and 50% of prospective single-arm studies. One-fourth of the studies did not report on preliminarily cessation of prophylaxis nor underlying reason. More worrisome was the fact that discontinuation of prophylaxis was mostly explained as adverse events and breakthrough infections, while patient and physicians' discretion as reasons were grouped together. On the other hand, we observed that only five studies did record infections that occurred after the prophylaxis had been stopped. This is concerning, as in a clinical setting and in developing guidelines, it is important to make a distinction of breakthrough infections during and after prophylaxis and pre-emptive therapy.[53] Also, it makes it more difficult to compare the efficacy of different medications,

for example when using ganciclovir and letermovir.[53] Furthermore, one of the included RCTs[17] reported early discharge from hospital as the main argument for stopping of prophylaxis. In addition, some reasons were not clarified, for example physician and patient discretion were combined in two of the included studies. A systematic review published in 2018 explored the efficacy and safety of CMV prophylaxis; adverse events and breakthrough infections were addressed, however the authors did not explicitly report additional information why patients stopped within these studies.[54] Similarly, in a PCP prophylaxis systematic review, the authors described adverse events as reason for discontinuation while not mentioning adherence, resistance or patients' choice.[14]

A systematic review about quality of reporting RCTs in medical oncology described that 79% of the adverse events in studies are reported according to Consolidated Standards of Reporting Trials (CONSORT) criteria, although the description of participants and preliminary stopping in each stage of the study was done correctly in only 59% of the studies.[55] It has been argued that perhaps, poor reporting is deliberate to mask the shortcomings in study design.[55 56] One may expect that in a prospective study, the patients are recruited and analysed prospectively thus the data about failure of prophylaxis like adverse events should be readily obtainable. Thus, registering

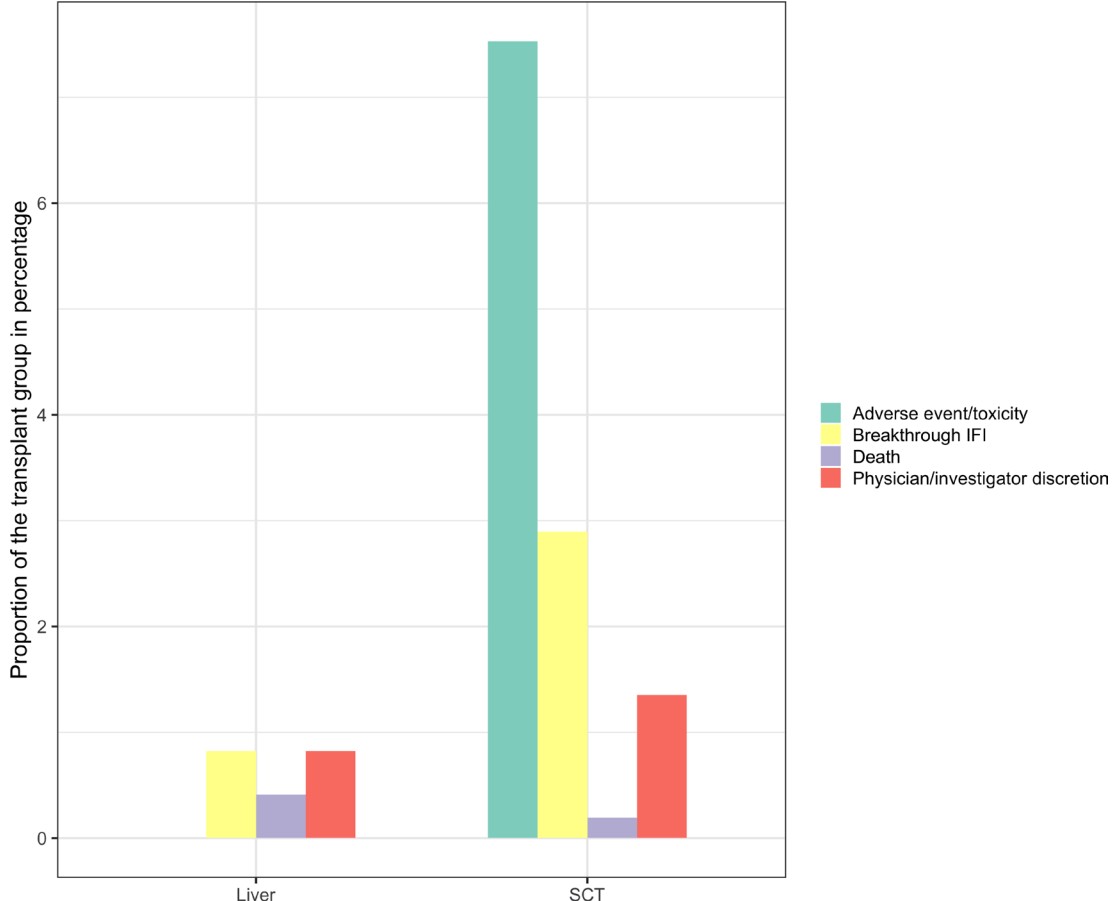

**Figure 3** Failure of IFI prophylaxis in five RCTs (y-axis represents the proportion of the transplant group in percentages and x-axis represents transplant). For one study, 15% of failure was caused by patients discharged early, this has not been included in this figure. IFI, invasive fungal infection.

study protocols in clinical trial registers and following reporting guidelines helps consistent and more straightforward reporting of results.[57 58] Goldacre and colleagues have looked into reporting of clinical trials specifically with regard to outcomes. They concluded that most of the studies did not have sufficient reporting—either outcomes were added or prespecified outcomes were not reported.[59] These results raise concern to also whether failure is reported as per protocol. In our case, we faced a substantial quantity of missing data on failure and it is not always clear to us how studies were conducted with respect to exclusion of patients. An additional statement to the CONSORT criteria regarding better reporting of harms in randomised trials is a useful guidance document to improve conduct of studies and address shortcomings.[60]

According to our search criteria, we also planned to include studies looking into prophylaxis and pre-emptive therapy of PCP, HHV-6 virus and febrile neutropenia. Surprisingly, we were not able to identify suitable studies looking into prophylaxis of these infections for our systematic review. Current guidelines for the treatment of these infections use case reports, retrospective studies, surveillance studies and outdated literature to give recommendations.[5 61 62] For instance, a systematic review focusing on PCP prophylaxis in non-HIV immunocompromised patients is used as main guidance of prophylaxis in this patient group; the studies included in this review date back from 1974 to 2008 (13 in the range 1977–1990).[14] Moreover, the prophylaxis of febrile neutropenia is widely supported by a systematic review that included studies with quinolones published from 1980 to 2010 (10 in the range 1980–1997).[13] Half of the studies in these reviews were published more than 20 years ago. Certainly, the evidence from these reviews are relevant to the field; however, the landscape of treatment of transplant recipients has changed—notably, with the emergence of resistant pathogens[63–66] and new data in today's setting is needed to aid the update of clinical guidelines.[62]

As mentioned before, detailed data about failure of prophylaxis and pre-emptive therapy were lacking in some studies and no additional information was obtained when reaching out to the authors. Furthermore, a limitation of our study was the restriction of start date (1 January 2010). This was done to avoid the effect of the significant change in management of infections and focus on the most recent evidence.[11 12] Another limitation is that we did not include studies about children. On the other hand, included studies already showed a heterogeneous variety of patient populations (eg, autologous and allogeneic SCT recipients),[19 48] various types of infections and

**Table 3** Risk of bias in RCTs using RoB tool

| | Randomisation process | Assignment of intervention | Adhering to intervention | Measurement of outcomes | Missing outcome data | Selection of the reported result |
|---|---|---|---|---|---|---|
| Humar *et al*, 2010[15] | + | + | + | + | + | ? |
| Halim *et al*, 2016[31] | ? | + | + | ? | + | ? |
| Witzke *et al*, 2012[33] | + | + | + | ? | + | ? |
| Reischig *et al*, 2012[35] | + | + | + | ? | + | ? |
| Reischig *et al*, 2015[36] | + | + | + | ? | + | ? |
| Togashi *et al*, 2011[38] | ? | + | + | ? | + | ? |
| Padulles *et al*, 2016[25] | + | + | + | + | + | ? |
| Palmer *et al*, 2010[27] | + | + | ? | + | + | ? |
| Chawla *et al*, 2011[28] | ? | ? | ? | ? | + | ? |
| Boeckh *et al*, 2015[29] | + | + | ? | + | + | ? |
| Kim *et al*, 2010[30] | ? | + | *+ | ? | + | ? |
| Perrella *et al*, 2012[47] | ? | ? | ? | ? | + | ? |
| Winston *et al*, 2014[17] | + | + | + | + | + | ? |
| Chaftari *et al*, 2012[48] | + | ? | ? | ? | + | ? |
| Huang *et al*, 2012[19] | + | + | ? | ? | + | ? |
| Park *et al*, 2016[49] | ? | + | + | ? | + | ? |

+low risk; ?, some concerns.
RCT, randomised controlled trials; RoB, risk-of-bias.

prophylaxis and pre-emptive therapy studies; therefore, we believe that adding paediatric studies would have further increased variability of our results.

There remains a variety of different practices between centres complicating patient transfers between hospitals.[9 67] We believe that having information about the discontinuation of failure of prophylaxis and pre-emptive therapy could

provide valuable information for guideline committees, medical practitioners and researchers conducting studies with these medications. Without this information, similar errors could be repeated in different studies.

There are several ways to predict failure in a clinical setting. Detecting adverse events and avoiding breakthrough infections can be done by approaching the

**Table 4** Risk of bias in single-arm studies using ROBINs tool

| | Confounding | Selection of participants | Classification of interventions | Deviations from intended interventions | Missing data | Measurement of outcomes | Selection of the reported result |
|---|---|---|---|---|---|---|---|
| Montejo *et al*, 2010[39] | – | ? | + | NI | – | ? | NI |
| Nanmoku *et al*, 2018[40] | * | + | + | + | + | ? | NI |
| Perrottett *et al*, 2009[41] | * | ? | + | ? | * | ? | NI |
| McGillicuddy *et al*, 2010[42] | * | + | + | * | * | ? | NI |
| Takenaka *et al*, 2012[45] | * | ? | + | ? | * | ? | NI |
| Park *et al*, 2012[46] | ? | + | + | ? | + | ? | NI |
| Mitsani *et al*, 2012[43] | * | ? | + | ? | ? | ? | NI |
| Cordonnier *et al*, 2010[44] | * | * | + | ? | ? | ? | NI |

+, low risk; ?, moderate risk; *, serious risk; -, critical risk.
NI, no information; ROBINS, Risk of Bias in Non-randomised Studies of Interventions.

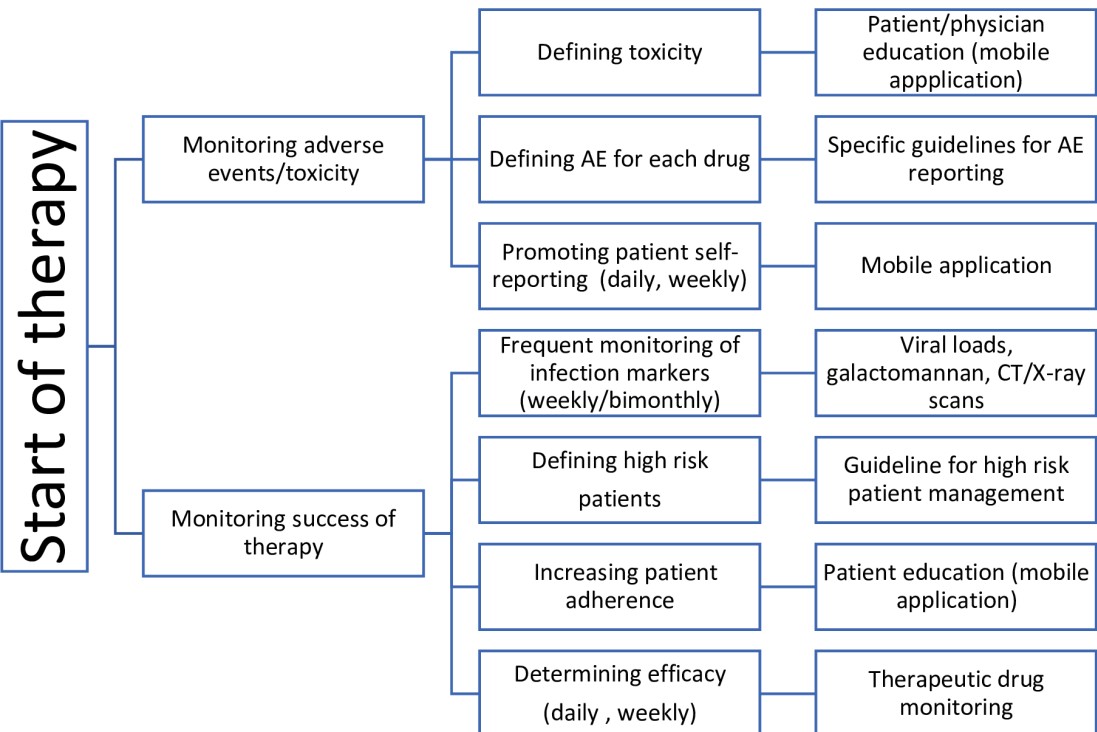

**Figure 4** Detecting failure in a clinical setting. AE, adverse event.

patient or treating the physician. For example, using mobile applications for reporting adverse events,[68] defining the high-risk patients[69 70] and therapeutic drug monitoring of potentially toxic medications.[71]

In figure 4, we have summarised guidance on how to detect failure early.[68–76] As mentioned above, there are multiple studies that do not report the true breakthrough infections, only postprophylaxis infections are reported. In addition, we believe that defining the potential adverse events before and describing the method of measurement of these could increase adherence. Other supporting background from the current systematic review for this figure is presented in Appendix 5.

## Conclusions

In general, RCTs and prospective single-arm studies about prophylaxis and pre-emptive therapy of opportunistic infections should provide more in-depth information about failure. The main reason why prophylaxis or pre-emptive therapy is stopped are adverse events; however, these may well be underreported. Thus, the management and reporting of adverse events is critically important and should be improved in clinical studies. In addition, our results suggest partially biased approach in the publication of clinical studies and therefore there are insufficient data to support evidence-based decision-making in prophylaxis of PCP, HHV-6 and febrile neutropenia.

## Author affiliations
[1]Department of Clinical Pharmacy and Pharmacology, University Medical Center Groningen, University of Groningen, Groningen, The Netherlands
[2]Department of Hematology, University Medical Center Groningen, University of Groningen, Groningen, The Netherlands
[3]Department of Gastroenterology and Hepatology, University Medical Center Groningen, University of Groningen, Groningen, The Netherlands
[4]Department of Pulmonary Diseases and Tuberculosis, University Medical Center Groningen, University of Groningen, Groningen, The Netherlands
[5]Department of Internal Medicine, University Medical Center Groningen, University of Groningen, Groningen, The Netherlands
[6]The University of Sydney, Sydney Pharmacy School, Sydney, New South Wales, Australia

**Acknowledgements** The authors would like to thank Mariska M.G. Leeflang for her suggestions and help while analyzing and compiling the data of this systematic review.

**Contributors** AGM did the screening, writing, data analysis, risk of bias analysis, searches and planning; MB did the screening, searches, data analysis, writing; HB, EAMV, SPB, LFRS and TSvdW did the planning, writing, reviewing of manuscript; JWA did the writing, analysis and planning.

**Funding** AGM was funded by Marie Skłodowska-Curie Actions, Grant Agreement number: 713660 — PRONKJEWAIL — H2020-MSCA-COFUND-2015

**Competing interests** None declared.

**Patient consent for publication** Not required.

**Provenance and peer review** Not commissioned; externally peer reviewed.

**Data availability statement** The data that support the findings of this study are available from the corresponding author.

**ORCID iDs**
Anne-Grete Märtson http://orcid.org/0000-0001-6478-1959
Jan-Willem C Alffenaar http://orcid.org/0000-0001-6703-0288

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
