## [Reviewer comments · BMJ Open]

ARTICLE DETAILS

TITLE (PROVISIONAL)	Exploring failure of antimicrobial prophylaxis and pre-emptive therapy for transplant recipients: a systematic review
AUTHORS	Mårtson, Anne-Grete; Bakker, Martijn; Blokzijl, Hans; Verschuuren, Erik; Berger, Stefan; Span, Lambert; van der Werf, Tjip S.; Alffenaar, Jan-Willem

VERSION 1 - REVIEW

REVIEWER	Jonathan Hand Ochsner Medical Center USA University of Queensland School of Medicine, Ochsner Clinical School
REVIEW RETURNED	29-Oct-2019

GENERAL COMMENTS	I applaud the authors for their interesting question. Attempts to glean and assess more granular details related to breakthrough and adverse drug events in these studies are warranted. Though these data reporting gaps are a known phenomenon in the literature, you have specifically highlighted the data gaps present in trials done in stem cell and solid organ transplant recipients. However there was not enough data reported in these trials to perform the intended analysis. I am not clear that the results address the primary and secondary research questions or objectives. the goal was to explore failures of antimicrobial prophylaxis and pre-emptive therapy - this was unable to be done given absence of data. Additionally, if a true systematic review is to be done, then why were no retrospective studies included? Additionally there is little to no mention of the findings or results related to resistance to antimicrobials, antifungals, or antivirals. Because specific data was not present it appears the focus of the manuscript shifted to a focus on absence of data and then a recommendation of how to detect failure early in a clinical setting. Again, this focus does not seem to answer the primary or secondary outcome. Another limitation is heterogeneity of populations and adverse drug events and breakthrough. Combining stem cell and solid organ transplant populations is problematic as they both experience different common complications (for instance GVHD, or surgical complications) and different drug-drug interactions. Also different drugs were evaluated that have very different spectra and known side effect profiles.
---

	Overall the manuscript is not focused and could be improved by better defining the goals and objectives and tying those more clearly to the results. For completeness I would also recommend evaluating retrospective studies if goal is systematic review.
--	---

REVIEWER	Martin Howell University of Sydney Australia
REVIEW RETURNED	11-Nov-2019

GENERAL COMMENTS	This systematic review addresses two important aspects. Firstly the subject of the review is a clinically important question to address. Secondly the review highlights the problems arising with inadequate reporting of harms in clinical trials. I have some comments to make that I hope might improve the manuscript.  1. The methods section should provide justification for the start date of 2010. 2. In the discussion section it is stated we therefore believe that: 'adding studies on children would have made it more difficult to summarize findings.' This is not a valid reason for not including children, particularly as the number of included studies is quite small. Children could have been addressed separately. Irrespective this should be addressed in the methods section not discussion. 3. Figures 2 and 3 show the number of participants. This should be shown as proportions of each of the transplant groups given the variation in total numbers across the groups. This is a particular problem as it is hard for the reader to find this number. To that end it would be useful either in the text or a summary table to give the split between transplant groups. As an example the 'other' numbers for kidney transplants are high, however the proportions may be similar, higher or lower when compared to the other groups. Other is problematic for the reason noted by the authors. 4. Given the above, suggested that 'other' be included in Table 2. 5. Table 2 is difficult to read. I suggest it be restructured to with reasons for stopping as columns. In addition the table should include total number for each trial and include proportions as well as n e.g. ## (##%). It is also unclear from table 2 where particular AEs have been specified and where they have not. Does the absence of brackets mean the AEs are unspecified. Do the inclusions in the brackets when provided refer only to AEs relevant to prophylaxis/pre-emptive treatment? Table 2 could provide more of this information. 6. In my view Table 3 does not provide any useful information and it does not contribute to the overall findings. Indeed only 2 sentences are given over to Table 3 in the results. The heterogeneity of the studies is clear from Table 1. That is they cover a range of interventions (different drugs, different doses, placebo etc.) as well as different populations. Furthermore, the events used to calculate ORs presumably combine undefined and defined events and events with very different aetiology. For example side effects, patient dropout, break through infection and so on. I recommend that Table 3 be removed. 7. I suggest that the authors refer to the CONSORT harms extension for minimum reporting requirements. Indeed this might have been a more useful guide to deficiencies of the included studies than the risk of bias assessment. Reference to CONSORT harms extension provides an objective basis for identifying where
---

	the studies are lacking and for recommendations of reporting requirements to address the problems. (Ioannidis JP, Evans S, Gotzsche P et al. Better reporting of harms in randomized trials: an extension of the CONSORT statement. Ann Intern Med 2004; 141: 781–788).
--	--

VERSION 1 – AUTHOR RESPONSE

Reviewers' Comments to Author:

Reviewer: 1

Reviewer Name: Jonathan Hand

Institution and Country: Ochsner Medical Center USA, University of Queensland School of Medicine, Ochsner Clinical School

Please state any competing interests or state 'None declared': none declared

I applaud the authors for their interesting question. Attempts to glean and assess more granular details related to breakthrough and adverse drug events in these studies are warranted. Though these data reporting gaps are a known phenomenon in the literature, you have specifically highlighted the data gaps present in trials done in stem cell and solid organ transplant recipients.

However there was not enough data reported in these trials to perform the intended analysis. I am not clear that the results address the primary and secondary research questions or objectives. the goal was to explore failures of antimicrobial prophylaxis and pre-emptive therapy - this was unable to be done given absence of data. Additionally, if a true systematic review is to be done, then why were no retrospective studies included? Additionally there is little to no mention of the findings or results related to resistance to antimicrobials, antifungals, or antivirals. Because specific data was not present it appears the focus of the manuscript shifted to a focus on absence of data and then a recommendation of how to detect failure early in a clinical setting. Again, this focus does not seem to answer the primary or secondary outcome. Another limitation is heterogeneity of populations and adverse drug events and breakthrough. Combining stem cell and solid organ transplant populations is problematic as they both experience different common complications (for instance GVHD, or surgical complications) and different drug-drug interactions. Also different drugs were evaluated that have very different spectra and known side effect profiles.

Overall the manuscript is not focused and could be improved by better defining the goals and objectives and tying those more clearly to the results. For completeness I would also recommend evaluating retrospective studies if goal is systematic review.

Response to reviewer 1:

- We did not include retrospective studies, because differences in documentation practises in medical records result in bias in retrospective studies. Philip Sedgwick has discussed some of the advantages and disadvantages of retrospective cohort studies, where he also mentions that in retrospective studies for example, the patient notes used can be incomplete (Sedgwick et al 2014 *BMJ*;348:g1072, doi: <https://doi.org/10.1136/bmj.g1072>).
- We defined the primary and secondary outcomes for this review beforehand to provide transparency and documented this in a protocol, which we uploaded to PROSPERO. The secondary outcome resistance to antimicrobials, antivirals and antifungals was usually

addressed in the introductions and discussions of the included studies, however not regarded as failure of therapy. Palmer et al 2010 and Boeckh et al 2015 (Table 1) both report on measurement of resistance, however in the study by Boeckh and colleagues no resistance was confirmed and Palmer and colleagues mention one known resistance mutation. Because a majority of all studies did not report on resistance and did not connect this to the failure of prophylaxis and/or pre-emptive therapy we did not include this in the review. The other secondary outcome, death we have reported in Table 2, however for clarity we have included a short section on this in the results section.

Revised section in results:

From the secondary outcomes death was reported more frequently – in 33% (n=8) of the studies death was the reason for failure^{17,27,30,31,35,36,44,46}. The secondary outcome resistance to antimicrobials, antivirals and antifungals was addressed in the introductions and discussions of the included studies, however not regarded as failure of therapy. Moreover, the presence and/or measurement of resistance to the study drug was described in 2 studies^{27,29}. Boeckh et al report no resistance genes in the investigated patients and Palmer et al report one patient with known resistance to ganciclovir.

- We included a wide population and different antimicrobials to see overview in different infectious diseases where routine prophylaxis and/or pre-emptive therapy is used, often for an extended period. As this is a burden for healthcare and as mentioned in our introduction and discussion – often not supported by new evidence, we believed that we could find patterns that could improve prophylaxis and/or pre-emptive therapy in the wider patient population. As mentioned before, we have included some objectives in the introduction.

Revised section in the introduction:

This systematic review will identify 1) causes of failure of prophylactic treatment, 2) factors that might contribute to failure of prophylaxis and 3) different approaches for administering prophylaxis.

This systematic review aimed to summarize the main reasons why prophylaxis and pre-emptive therapy has failed in solid organ and allogeneic stem cell transplant recipients and how failure is reported in prospective studies. In this review we focus on failure of prophylactic therapy, during the treatment window.

Reviewer: 2

Reviewer Name: Martin Howell

Institution and Country: University of Sydney, Australia

Please state any competing interests or state 'None declared': None declared

This systematic review addresses two important aspects. Firstly the subject of the review is a clinically important question to address. Secondly the review highlights the problems arising with inadequate reporting of harms in clinical trials. I have some comments to make that I hope might improve the manuscript.

1. The methods section should provide justification for the start date of 2010.

Response to reviewer:

We have provided this justification in the discussion section:

Furthermore, a limitation of our study was the restriction of start date (1st January 2010). This was done to avoid the effect of the significant change in management of infections and focus on the most recent evidence^{11,12}.

However, we will include a statement also in the methods section.

Revised section in methods:

This systematic review included prospective randomized controlled trials and prospective single-arm studies from January 1st 2010 to October 1st 2018. The starting date was January 1st 2010 due to the changes in management of different infections in recent decades and to include the most recent evidence^{11,12}.

2. In the discussion section it is stated we therefore believe that: 'adding studies on children would have made it more difficult to summarize findings.' This is not a valid reason for not including children, particularly as the number of included studies is quite small. Children could have been addressed separately. Irrespective this should be addressed in the methods section not discussion.

Response to reviewer:

We agree, that this study could have indeed included children. However, as we already have quite a heterogeneous patient population with multiple infections being investigated, we believe that children introduce even more variability and are not the scope of this review. We have included a statement in the methods section.

Revised section in methods:

The review included patients (16 years and older) who had received either allogeneic stem cell, lung, kidney, liver, heart, pancreas or small bowel transplantation. Studies done on children under 16 years were not included as this would have introduced more variability and require a separate analysis, thus were out with the scope of this review.

Revised section in discussion:

.. various types of infections and prophylaxis and pre-emptive therapy studies; therefore, we believe that adding paediatric studies would have further increased variability of our results.

3. Figures 2 and 3 show the number of participants. This should be shown as proportions of each of the transplant groups given the variation in total numbers across the groups. This is a particular problem as it is hard for the reader to find this number. To that end it would be useful either in the text or a summary table to give the split between transplant groups. As an example the 'other' numbers for kidney transplants are high, however the proportions may be similar, higher or lower when compared to the other groups. Other is problematic for the reason noted by the authors.

Response to reviewer:

Yes, we agree, for clarity we have changed the figures to proportions in percentages, for the IFI group in one particular study (Winston et al 2014), a large proportion of the whole group (15%) has a failure reason: discharge from hospital. We mentioned this in the legend of the figure to have better clarity on the figure for other reasons. We believe that having the proportion in the y-axis makes it easier to follow the number.

Revised figures:

Figure 2. Failure of CMV prophylaxis and pre-emptive therapy in 11 RCTs (y-axis represents the proportion of the transplant group in percentages and x-axis represents transplant).

Figure 3. Failure of IFI prophylaxis in 5 RCTs (y-axis represents the proportion of the transplant group in percentages and x-axis represents transplant). For one study¹⁷, 15% of failure was caused by patients discharged early, this has not been included in this figure.

4. Given the above, suggested that 'other' be included in Table 2.

Response to reviewer:

The 'other reasons' has been reported in Table 2, we have restructured Table 2 (see point 5).

5. Table 2 is difficult to read. I suggest it be restructured to with reasons for stopping as columns. In addition the table should include total number for each trial and include proportions as well as n e.g. ## (##%). It is also unclear from table 2 where particular AEs have been specified and where they have not. Does the absence of brackets mean the AEs are unspecified. Do the inclusions in the brackets when provided refer only to AEs relevant to prophylaxis/pre-emptive treatment? Table 2 could provide more of this information.

Response to reviewer:

Thank you, it is important indeed to improve clarity of this table. We have restructured Table 2.

Revised Table 2:

Table 2. Reasons for stopping prophylaxis or pre-emptive therapy

Study	AE	Breakthrough infection	Graft loss	Death	Patient decision	Physician/ investigator decision	Other	Total number subjects
Cytomegalovirus prophylaxis/pre-emptive therapy								
Humar et al, 2010	27 (8%)	51 (16%)			11 (3%)		14 (4%)	320
Halim et al, 2016			2 (1%)	1 (0.5%)				201
Reischig et al 2012			10 (14%)	2 (3%)				70
Reischig et al 2015[1]	13 (11%)		4 (3%)	3 (2.5%)				119
Padulles et al 2016	1 (2%)							53
Palmer et al 2010	17 (12.5%)			7 (5%)	3 (2%)	15 (11%)	3 (2%)	136
Boeckh et al 2015	44 (24%)	35 (19%)				20 (11%)		184
Kim et al 2010	1 (1.5%)			2 (3%)				68
Montejo et al 2010		3 (9%)						34
Perrottet et al 2009		3 (viremia)						65
McGillicuddy et al 2010	3 (2%)	1 (1%) [2]						130
Park et al 2012 [3]	8 (8%)	6 (6%)		37 (38%)				97
Invasive fungal infection prophylaxis								
Winston et al 2014 [4] Fluconazole		2 (2%)		1 (1%)		2 (2%)		100 (fluconazole) 200 (total)
Chaftari et al 2012 Posaconazole	8 (38%)							21 (posaconazole) 40 (total)
Huang et al 2012 Itraconazole	29 (20%)	12 (8%)		1 (1%)		4 (3%)		147 (itraconazole) 228 (total)
Park et al 2016 Fluconazole	2 (2%)	3 (3%)				3 (3%)		89 (fluconazole) 250 (total)

Mitsani et al 2012	25 (27%)			93
Cordonnier et al 2010	2 (4%)	3* (7%)	11* (2%)	45
NI in the following trials (timepoints not specified)				
Nanmoku et al 2018				100
Chawla et al 2011				37
Witzke et al 2012				296
Perrella et al 2012				43
All patients completed treatment protocol in the following trials				
Togashi et al 2011				22
Takenaka et al 2012				20

AE – adverse event, NI – no information (not reported), IFI – invasive fungal infection, CMV – cytomegalovirus

* time-point not confirmed

1. 1 patient moved to another country
2. No information on breakthrough infections during prophylaxis
3. 12 patients had dosage reductions
4. 36 patients were preliminarily stopped due to discharge from hospital

6. In my view Table 3 does not provide any useful information and it does not contribute to the overall findings. Indeed only 2 sentences are given over to Table 3 in the results. The heterogeneity of the studies is clear from Table 1. That is they cover a range of interventions (different drugs, different doses, placebo etc.) as well as different populations. Furthermore, the events used to calculate ORs presumably combine undefined and defined events and events with very different aetiology. For example side effects, patient dropout, break through infection and so on. I recommend that Table 3 be removed.

Response to reviewer:

We agree, as the heterogeneity is clear from other sources, we have decided to remove Table 3. The outcomes are difficult to compare due to difference in patient population and study medications.

7. I suggest that the authors refer to the CONSORT harms extension for minimum reporting requirements. Indeed this might have been a more useful guide to deficiencies of the included studies than the risk of bias assessment. Reference to CONSORT harms extension provides an objective basis for identifying where the studies are lacking and for recommendations of reporting requirements to address the problems. (Ioannidis JP, Evans S, Gotzsche P et al. Better reporting of harms in randomized trials: an extension of the CONSORT statement. *Ann Intern Med* 2004; 141: 781–788).

Response to reviewer:

Thank you, this is a good recommendation, these reporting requirements are very useful and important to follow.

Revised section in discussion:

An additional statement to the CONSORT criteria regarding better reporting of harms in randomized trials is a useful guidance document to improve conduct of studies and address shortcomings⁶¹.

VERSION 2 – REVIEW

REVIEWER	Martin Howell University of Sydney Australia
REVIEW RETURNED	02-Dec-2019

GENERAL COMMENTS	The authors have considered and addressed my suggestions. I look forward to publication of the manuscript.
--